# New Data on Spermatogenic Cyst Formation and Cellular Composition of the Testis in a Marine Gastropod, *Littorina saxatilis*

**DOI:** 10.3390/ijms21113792

**Published:** 2020-05-27

**Authors:** Sergei Iu. Demin, Dmitry S. Bogolyubov, Andrey I. Granovitch, Natalia A. Mikhailova

**Affiliations:** 1Institute of Cytology of the Russian Academy of Sciences, 4 Tikhoretsky Ave., 194064 St. Petersburg, Russia; natmik@mail.ru; 2Department of Invertebrate Zoology, St. Petersburg State University, 7/9 Universitetskaya Emb., 199034 St. Petersburg, Russia; granovitch@mail.ru

**Keywords:** testes, spermatogenesis, spermatocysts, cyst cells, marine snails, *Littorina saxatilis*

## Abstract

Knowledge of the testis structure is important for gastropod taxonomy and phylogeny, particularly for the comparative analysis of sympatric *Littorina* species. Observing fresh tissue and squashing fixed tissue with gradually increasing pressure, we have recently described a peculiar type of cystic spermatogenesis, rare in mollusks. It has not been documented in most mollusks until now. The testis of adult males consists of numerous lobules filled with multicellular cysts containing germline cells at different stages of differentiation. Each cyst is formed by one cyst cell of somatic origin. Here, we provide evidence for the existence of two ways of cyst formation in *Littorina saxatilis*. One of them begins with a goniablast cyst formation; it somewhat resembles cyst formation in *Drosophila* testes. The second way begins with capture of a free spermatogonium by the polyploid cyst cell which is capable to move along the gonad tissues. This way of cyst formation has not been described previously. Our data expand the understanding of the diversity of spermatogenesis types in invertebrates.

## 1. Introduction

Gastropod mollusks are key species in many marine, freshwater and land communities. They serve as the first intermediate hosts for trematodes and in this way, they support natural foci of epizootic diseases including important for man and domestic animals [1,2]. The data of the population/parasitological analysis of littorinids are presented in a number of publications [3,4,5,6]. *Littorina saxatilis* (Olivi, 1792) is an intermediate host for 12 species of trematodes belonging to eight families, they all cause parasitic castration of the infected snails. The definitive hosts of these parasites are seabirds and fishes. Thus, consideration of the reproduction of mollusks, including patterns of spermatogenesis, is important from both a population and ecosystem point of view.

Together with its related species, *L. saxatilis* is a model object for studying mechanisms of speciation and reproductive isolation. It is also a promising species because a sequence database of the *L. saxatilis* genome is now available [7]. However, it is surprising that the morphologic features of spermatogenesis and the cellular composition of male gonads in this species have not been exhaustively studied, despite the fact that such knowledge is important for solving many problems of evolutionary biology.

Spermatogenesis is a multi-stage, complex and finely tuned process, which includes a sequence of various morphologic, cytological and physiological events. It characterizes all sexually propagating animals and leads to the producing of a huge number of haploid spermatozoa from a small amount of diploid spermatogonial stem cells [8,9]. The mitotically dividing cells—spermatogonia—enter meiosis and give rise to spermatocytes. Meiosis is a fundamental event underlying reproductive success. The crucially important period of meiosis is meiotic prophase, during which primary spermatocytes successively pass through leptotene, zygotene, pachytene, diplotene and diakinesis stages. Two hallmark evens of meiosis—homologous recombination and pairing of the homologous chromosomes—occur in meiotic prophase. Meiosis includes two subsequent divisions—reductive and equational—with no intervening DNA synthesis [10,11]. As a result, after completion of meiosis, secondary spermatocytes become haploid spermatids that further differentiate into mature spermatozoa during spermiogenesis process [8].

Despite this general scheme, the mode of spermatogenic cell development depends on the testis structure and specific structural and functional relations between the germline cells and various specialized auxiliary somatic cells of the gonad [8]. There is no single nomenclature to designate the types of spermatogenesis and testes, which is primarily determined by traditions established in zoological literature; this is also true for Mollusca.

Two major types of spermatogenesis are cystic and tubular, both mostly studied in vertebrates [12,13,14]. Since the germ cells are distributed within the seminiferous tubule in both spermatogenesis modes, the tubular type of spermatogenesis can also be referred to as non-cystic (acystic) [14] to stress significant microanatomic and hence physiological differences between these two general spermatogenesis modes.

The cystic mode of spermatogenesis implies that germline cells develop within a cyst (spermatocyst)—the basic functional unit of the gonad. The first way for the cyst to form is by cytoplasmic extensions of the specialized auxiliary cells of somatic origin. These cells are located in the seminiferous epithelium and in anamniotic vertebrates [13,15], called the Sertoli cells (SCs). The second way of cyst formation is by special somatic cells called the cyst cells, e.g., in *Drosophila* [16]. The *Drosophila* testis contains two types of stem cells—germline stem cells and cyst stem cells—which give rise to the germline cells and cyst cells, respectively. In *Drosophila*, the cytoplasmic extensions of two cyst cells enclose a single goniablast, forming a cyst [17]. The goniablast then generates a clone of synchronously developing germ cells in this cyst. This spermatogenesis mode may be rather efficient in supporting germ cell development, since cyst cells concentrate specific growth factors required for each developmental phase. Alternatively, each non-cystic SC in the seminiferous epithelium at tubular spermatogenesis simultaneously supports different developmental stages of germline cells that are arranged in a developmental gradient, which helps to realize complex requirements for nursing different germ cell clones at the same time [12]. In vertebrates, the cystic type of spermatogenesis is ancestral [14] and characterizes the representatives of Anamnia—fish [12,18] and amphibians [19,20]. The tubular type of spermatogenesis is a characteristic feature of Amniota—reptiles [21], birds [22] and mammals [23].

Due to the association of spermatogonial stem cells with SCs, local niches of the gonad are formed. In these niches, a specific microenvironment determines the self-renewal of spermatogonial stem cells and their further differentiation [24]. The processes of germline cell differentiation are highly synchronized within each local niche due to incomplete cytokinesis of dividing descendants of a single spermatogonial stem cell. As a result, a clone of germline cells that are at the same stage of development is formed in the niche. This is an evolutionarily conserve feature of spermatogenic cell division [14]. In different niches of the gonad, spermatogonial stem cells enter differentiation asynchronously and hence the adjacent germ cell clones are usually at different stages of spermatogenesis. A key component of the local gonad niches is testicular endothelial cells that produce the necessary factors for self-renewal of spermatogonial stem cells [25].

In the seminiferous tubule, each non-cystic SC supports the development of several germline cell clones that are arranged in the following sequence, forming a differentiation gradient from the wall of the tubule to its lumen: spermatogonia, primary spermatocytes, secondary spermatocytes, early spermatids, spermiogenic spermatids and mature spermatozoa. In the case of Anamnia, each testicular cyst contains a group of synchronously developing descendants of a single spermatogonium. These cysts are formed by the cytoplasmic processes of SCs, which completely envelope the spermatogonium, and further differentiation of germline cells occurs within these spermatocysts. The SCs of Anamnia are homologous to the SCs of Amniota, but the former are able to proliferate even in adult animals [26].

The cystic type of invertebrate spermatogenesis and the relationships between the somatic cells and developing germline cells are the best studied in *Drosophila* [16,17,27,28,29]. However, *Drosophila* cystic spermatogenesis has special features that distinguish it from a cystic spermatogenesis of vertebrates, such as bony fishes and other Anamnia. In *Drosophila*, cysts form in the apical part of the testis, where the local gonadal niche—the hub—is located. This hub contains both cyst progenitor somatic cells and germline stem cells [30]. A somatic stem cell undergoes asymmetrical mitosis, as a result of which one daughter cell renews the stem cell pool and the second proceeds to spermatogenic differentiation that ends with formation of a goniablast cyst [17,31]. Each goniablast cyst in *Drosophila* includes two cyst cells of somatic origin and a goniablast, the germline cell [32]. The goniablast cysts detach from the niche and enter the lumen of the testicular tube. The differentiation gradient of germline cells is observed in the direction from the apical to the distal end of the *Drosophila* testis. As a result, one can distinguish the following successive zones enriched in the cysts that contain germline cells at the same stage of differentiation: spermatogonia, primary spermatocytes, secondary spermatocytes, spermatids during spermiogenesis and mature spermatozoa.

We have recently found that spermatogenesis of the rough periwinkle *Littorina saxatilis*, a marine gastropod, has some surprising similarities with spermatogenesis of *Drosophila* rather than most mollusks [33]. A detailed study of the structure and cellular composition of the *L. saxatilis* testis had not been carried out before the beginning of that study, despite the fact that there are detailed ultrastructural descriptions of developing male germline cells [34] and mature spermatozoa [35,36] in several species of *Littorina*.

In the present work, we discuss some additional aspects of anatomic and cellular organization of the *L. saxatilis* testis. We provide evidence for the existence of specialized somatic cells in the *L. saxatilis* testis wall, referred here as Sertoli-like cells (SLCs). The anastomosing cytoplasmic processes of the SLCs protrude deeply to the lobule cavity, but never form cysts as in a bivalve, *Margaritifera laevis* [37]. The spermatocysts of *L. saxatilis* originate from other somatic cells—the cyst cells. We described two independent differentiation ways of these spermatogenic cysts and demonstrated that one of them somewhat resembles the development of *Drosophila* cysts, whereas the other has never been described previously. In the latter case, the nucleus of a free cyst cell undergoes endoreduplication. These cyst cells are able to move and capture spermatogonia. In sum, our data additionally characterize the specific lobular-cystic type of the *L. saxatilis* testis has never been described in gastropods before.

## 2. Results

The unpaired testis of *L. saxatilis* is firmly embedded in the digestive gland termed the hepatopancreas (Figure 1A) and both organs are located in a fascial bag. The *L. saxatilis* testis has a lobular architecture and the blind-ended lobules are basic anatomic units of the gonad. To avoid confusion in the anatomy studying of the *Littorina* testis, it is necessary to remove the wall of the hepatopancreas and then flatten the fresh gonad on a slide. The lobules differ from each other in shape, size and color within the same testis, which depends on the degree of development (Figure 1B).

To further study the cellular composition of the lobules, we applied a squashing technique using a slow pressure increase [38]. This method has proven to be useful for high-resolution cytological observations without visible cell damage [33]. As a result, we were able to identify spermatocysts as morphologically integral entities of the *L. saxatilis* testis.

The wall of the lobules in the *L. saxatilis* testis contains large somatic cells, referred here to as the SLCs to distinguish them from typical SCs. These are flattened polygonal cells, whose cytoplasm is packed with characteristic yellow-colored granules, scarce in young and extremely numerous in terminally differentiated cells (Figure 2 and Figure 3A, respectively). Adjacent SLCs are always contact closely with each other. Small agranular cells that most likely represent the SLC progenitors are also observed (Figure 2). Mature SLCs have enormous size. Their long cytoplasmic extensions permeate deeply inside the lobule and can easily be identified by the presence of pigment granules (Figure 3A).

The main feature of the *L. saxatilis* testis lobules are numerous distinct spermatogenic cysts (spermatocysts) that contain germline cells at different stages of differentiation, but each cyst possesses a clone of the same stage cells only (Figure 3B,C). A few free spermatozoa can also be observed in the lobule lumen.

In those testis lobules, in which spermiation generally completes or is at the terminal stage, the processes of the late SLCs are anastomosed, forming a loose branching meshwork that reinforces the lobule cavity (Figure 4A,B). Rare remaining cysts with spermatids and even secondary spermatocytes can sometimes be observed in the SLC mesh (Figure 4B).

We can identify two independent lines of *L. saxatilis* cyst differentiation. The first one (Figure 5A–C), partially described by us earlier [33], resembles the line of cyst differentiation in *Drosophila* testes [29,39]. In this case, the cyst begins to differentiate with the join of a stem cyst cell and goniablast, forming a goniablast cyst. Such cysts are formed due to the ability of somatic cyst cells to grow continuously with simultaneous mitotic blockade. Conversely, the germline cells are incapable of continuous growth, but acquire the ability to divide mitotically and then entry meiosis. Noticeably, the spermatogonial, spermatocytic and spermatid cysts of *L. saxatilis* are always formed by a single cyst cell (Figure 5A–C). During differentiation of these cysts, chromatin of the cyst cell progressively compacts (Figure 5B). At the end of cyst differentiation, the nucleus of cyst cells degrades (Figure 5C,D) and may not be visible in the cysts containing mature sperm.

An alternative line of *L. saxatilis* spermatocyst differentiation was discovered by us for the first time (Figure 6 and Figure 7). It is associated with the coexistence of free cyst cells and free spermatogonia in the gonad. In the absence of a contact with spermatogonia, free cyst cells become capable of endoreduplication. The examples of these free cyst cells at successive stages of polyploidization are represented in Figure 6. Judging by the size of the nuclei and the cells themselves, free cyst cells can undergo at least five to six rounds of endoreplication. Their nucleus contains a significant amount of diffuse chromatin (Figure 6F–K) and several compact nucleoli (Figure 6L–O), the number and size of which increase in proportion to the level of cell ploidy.

Using a chamber for vital observation, we saw that free cyst cells can form lobopodia (Figure 6B,C) and move along the surface of the chamber in this manner. This apparently allows free cyst cells to capture free spermatogonia (Figure 7b) and form specific cysts that can be easily distinguished by a polyploid nucleus of the cyst cell (Figure 7A–C). In the testis of sexually mature *L. saxatilis* males, this second line of cyst differentiation is noticeably more common. In this mode of cyst development, the cyst cells do not necessarily degrade after sperm excystation since we have repeatedly observed “empty” cysts with no signs of chromatin degradation in the cyst cell nucleus.

## 3. Discussion

We have recently reported that the rough periwinkle, *L. saxatilis*, demonstrates some peculiar features of the testis structure never described earlier in other mollusks [33]. The *L. saxatilis* testis has a lobular-cystic architecture and cannot be referred to as a tubular testis, as suggested previously [40]. It consists of blind-ended lobules packed with multicellular spermatogenic cysts. An individual cyst is formed by a single cyst cell of somatic origin. Each cyst contains 1–128 germline cells at the same stage of differentiation. In addition, bicellular goniablast cysts, single spermarmatogonia, diploid and polyploid somatic cells (free cyst cells) and excysted mature spermatozoa can also be found in the lobule lumen of the *L. saxatilis* adult testis [33].

In parallel with our studies, the authors of [37] described another cystic mode of spermatogenesis in a bivalve mollusk, *Margaritifera laevis*. However, the mode of *M. laevis* cystic spermatogenesis, unlike *L. saxatilis*, somewhat resembles spermatogenesis of bony fishes. In the *M. laevis* testis, the clones of synchronously developing spermatogenic cells are completely surrounded by the cytoplasmic processes of somatic cells, referred by the authors to as SCs [37]. It has previously been suggested that any type of cystic spermatogenesis is not characteristic for mollusks. Ironically, the mode of cystic spermatogenesis observed in *Littorina* formally resembles the mode of *Drosophila* spermatogenesis [17,32] rather than of other mollusks.

We showed in the present study that the spermatogenic cysts of *L. saxatilis* testes can form at least in two independent ways. In the first, primary cysts appear due to an association of a somatic cyst cell with a goniablast. The formation of primary (goniablast) cysts at the periphery of local spermatogonial stem cell niches is inherent to the testis of *D. melanogaster* [29,39]. In the second, free spermatogonia are captured by free cyst cells that underwent several rounds of endoreduplication. This mode of cyst formation seems a specific feature of *L. saxatilis* spermatogenesis.

### 3.1. Do SLCs of the L. saxatilis Testis Resemble True SCs?

Spatial and functional relations between somatic and germline cells in molluskan testes remain largely unknown despite of numerous ultrastructural studies of spermatogonia, spermatocytes, spermatids and especially mature spermatozoa, carried out in many bivalves and gastropods, including several species of *Littorina* [34,35,36]. The seminiferous epithelium of both cystic and tubular testes is characterized by the presence of specialized somatic cells termed the Sertoli cells (SCs) in vertebrates [13,15]. The presence of similar cells has been documented for many mollusks. These cells were described under different names, but more often as the SCs [37,41,42,43,44]. These SCs (or Sertoli-like cells, SLCs) primarily play a structural/accessory role for the developing spermatogonial cells and probably also involved in steroid hormone production, at least in *Mytilus* [44]. However, it is still debatable whether such cells resemble vertebrate SCs in all respects.

We found that the *L. saxatilis* testis contains conspicuous somatic cells referred here as the SLCs. In our opinion, the *L. saxatilis* SLCs bear little resemblance to the SCs observed in the tubular testes of vertebrates, as suggested previously for *L. sitkana* [41]. By location in the gonad, shape and especially due to the numerous pigment inclusions in the cytoplasm, these SLCs somewhat resemble the pigment cells of *D. melanogaster* testes [45]. It should be noted that *Drosophila* pigment cells are surprisingly poorly studied in the functional and molecular aspects. It is only known that they play a barrier role and are necessary to develop sexual dimorphism in flies [46].

Buckland-Nicks and Chia [41] studied the ultrastructure of SLCs in *L. sitkana* and two other marine snails, *Ceratostoma foliolatum* and *Fusitriton oregonensis*. In the last two species, these cells are really very similar to vertebrate SCs. However, the only illustration of such a cell from the *L. sitkana* testis presented in the work cited is insufficient to compare this cell with *L. saxatilis* SLCs or with the SCs of vertebrates with their tubular-type gonads.

Unlike the tubular mode of spermatogenesis demonstrating no cysts in the testis—this mode is typical for most molluskan species—a cystic mode has not been described in any mollusk until recently. Kobayashi and co-authors [37] were the first who have documented that the cytoplasmic processes of somatic cells resembling SCs in the testis of the freshwater pearl mussel, *Margaritifera laevis*, can totally envelope the clones of germ cells to form spermatocysts similar to those in the testes of teleost fishes. The *L. saxatilis* SLCs also exhibit extremely long cytoplasmic protrusions that, however, never form spermatocysts and thus these cells cannot be regarded to as the counterparts of vertebrate SCs.

Interestingly, another bivalve mollusk, *Mytilus galloprovincialis*, demonstrates the specific structure of the seminiferous tubule, with peculiar follicles containing spermatogenic cells at various stages of differentiation, including mature spermatozoa [44]. However, the mode of *Mytilus* spermatogenesis is apparently not cystic. The data presented in the study [44] suggest that there is a special microanatomic structure of the *Mytilus* seminiferous tubule, which can be considered as a result of invagination of the basement membrane into the tubule lumen. This is accompanied by the formation of follicles that exhibit a prominent cavity. Neither the cavity nor the basement membrane can be observed in the spermatogenic cysts.

### 3.2. Two Ways of Cyst Formation in the L. saxatilis Testis

*L. saxatilis* male germline cells develop within peculiar cysts formed by somatic cyst cells. In the present work, we were able to establish at least two independent ways of cysts formation in the testis of this species. The first way involves the formation of a goniablast cyst on the periphery of the local stem cell niche of the testis, which is also inherent in *Drosophila* [29,39]. The differences between the goniablast cysts of *Littorina* and *Drosophila* concern the number of somatic cells forming the cyst: two in the fly and only one in the mollusk. These differences determine the orientation of germ cells in the cyst in the late stages of spermatogenesis. In *Drosophila*, germline cells are strictly ordered, starting from the stage of spermatid elongation. In this case, a collateral bunch of spermatids or spermatozoa forms in the cyst, with the same orientation of the tails and acrosomal ends. At the ascertain stages of *L. saxatilis* spermatogenesis, the acrosomal and tail ends of different spermatids/spermatozoa are oriented randomly in an individual cyst [33]. This is especially clearly seen in the cysts at the terminal stages of spermatogenesis, when individual germ cells are spatially separated from each other, as a rule.

Like in *Drosophila*, the goniablast cysts of *L. saxatilis* are able to migrate into the gonad lumen. During the spawning season, the goniablast cysts of *L. saxatilis* are rarely observed at the periphery of the apical parts of testis lobules and were found upon by chance only in our previous work [33]. A screening of local stem cell niches in *L. saxatilis*, where the goniablast cysts may locate, meets technical difficulties, because of the large size of male gonads and the absence of molecular stem cell markers.

The second way of *L. saxatilis* spermatocyst formation involves the capture of free spermatogonia by free cyst cells (Figure 5), whose nuclei undergo several rounds of endoreduplication (Figure 6). This mode of cyst formation and the ability of cyst cells to migrate had not yet been described and is likely a specific spermatogenesis feature of periwinkles. The cytochemical characteristics of free polyploid cyst cells suggest a high level of their functional activity.

Noticeably, free cyst cells significantly differ in morphology from the SLCs found both in the gastropod *F. oregonensis* [41] and in the bivalve *M. laevis* [37]. These cells are also not similar to free cyst cells registered in *D. melanogaster* agametic mutants [32], in which mutations of the male fertility genes *chickadee* and *diaphanous* result in a blockade of reproduction and differentiation of the germline stem cells. In such a case, agametic testes of the mutants are filled with clusters of mitotically dividing cyst cells. The mitotic activity of free somatic cyst cells in *Drosophila* mutants brings them closer to the SCs of vertebrates with the cystic testes, whose SCs remain mitotically active [26], unlike terminally differentiated SCs in the tubular-type testes [13].

Interestingly, the differentiation of *L. saxatilis* cysts formed from a goniablast cyst is accompanied by progressive chromatin compaction in the cyst cell nucleus and its gradual degradation in the end of spermatogenesis (Figure 5). The cysts of this origin are completely degraded after excystation of spermatozoa. With the alternative genesis of cysts, the endoreduplicated nucleus of free cyst cells does not show signs of degradation not only at the terminal stages of spermatogenesis (Figure 7C), but even after the sperm release.

In conclusion, one must be conscious of the single name “cystic spermatogenesis” can actually combine different modes of germ cell development. The obtained new data on *L. saxatilis* spermatogenesis will allow us to understand better the mechanisms of reproductive isolation, the reasons for gamete incompatibility and evaluate the role of gamete interaction proteins in the context of the origin of *Littorina* sibling species and complexes of its sister species in nature. Some data have recently been published on this topic [47,48,49], but many questions still remain open.

## 4. Materials and Methods

Sexually mature males of *L. saxatilis* (Olivi, 1792) were collected in spring, during the spawning season of this species, at the littoral zone of the Barents Sea at Tromsø Island, Norway. Males infected with any kind of trematode were excluded from the analysis due to the destruction of reproductive traits and parasitic castration of the penis and testis. The mollusks were kept in aquaria with running sea water at 10 °C and an automatic imitation of the tidal cycle. The study was approved by the Animal Ethics Committee of the Institution of Cytology of the Russian Academy of Sciences (Assurance Identification number F18-00380).

After removal of the shell, the primary preparation of the gonads was carried out on a glass slide in a drop of filtered seawater limited by strips (5 × 26 mm) of adhesive tape. Further fine preparation was carried out in 120 μL of quail protein, after which the preparation was carefully covered with a coverslip and lightly pressured. During this, an adhesive tape formed a chamber, allowing microscopic observations of unfixed cells for 2–3 h. The cells were observed under a Leica DM 2500 microscope equipped with Nomarski optics (DIC) and a DFC 420 CCD camera using a 100 × immersion objective (NA 1.25).

For cytological analysis, squashed preparations were prepared using the high pressure technique developed by us earlier [38]. The testes were incubated in hypotonic medium (130-mM KCl) for 20 min at room temperature, then fixed with methanol and glacial acetic acid (3:1) for 2–4 h at room temperature and stored at −20 °C before use. The fixed testes were divided into small fragments in 50% propionic acid, then covered with a siliconized coverslip (24 × 24 mm), after which the preparations were placed in a vertical hydraulic vise equipped with a manometer and pressure was gradually increased to about 250 kg/cm^2^ for 90–120 s. The preparations were frozen in liquid nitrogen, after which the coverslip was removed with a razor blade, dehydrated in an ascending series of ethanol (70%, 80% and 100%), dried in air and stored at −20 °C.

A portion of preparations was stained with silver nitrate according to the standard AgNOR procedure [50]. Chromomycin A_3_ (CMA_3_) staining was performed according to Schweizer [51], with some modifications. In this case, stock solution of CMA_3_ (Sigma-Aldrich) was prepared in deionized water by dissolving 1 mg/mL CMA_3_ for several days in the darkness without stirring at 4 °C. Working solution of CMA_3_ (0.5 mg/mL) was prepared in McIlvaine buffer solution (pH = 7.0) containing 5-mM MgCl_2_. After rinsing in the same buffer, the preparations were stained with CMA_3_ for 1–2 h in the darkness at room temperature and mounted in a ProLong^®^ Gold antifade medium (Invitrogen). The stained preparations were left in darkness for 3–5 days at 4 °C to stabilize CMA_3_ fluorescence and then imaged with a Leica DMI 6000 inverted microscope.

## Figures and Tables

**Figure 1 ijms-21-03792-f001:**
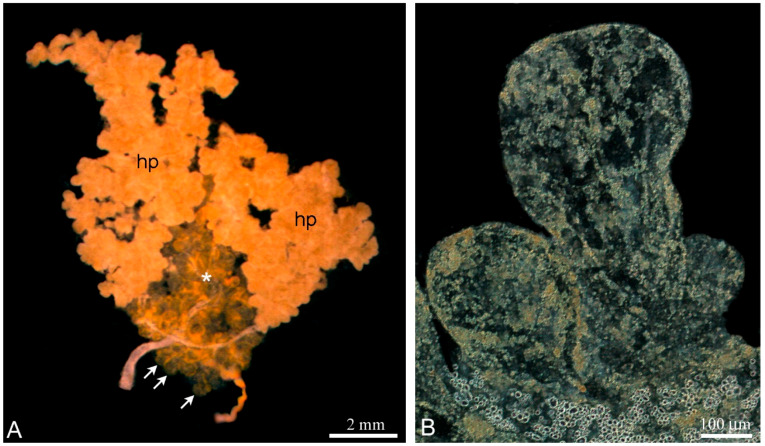
Anatomy of the *Littorina saxatilis* testis; (**A**) Disposition of the testis (*asterisk*) in the isolated unfixed hepatopancreas (hp) flattened on the slide. The hepatopancreas acini that cover the testis are dissected and partially removed. Arrows indicate individual lobules of the testis; (**B**) adjacent testicular lobules at higher magnification.

**Figure 2 ijms-21-03792-f002:**
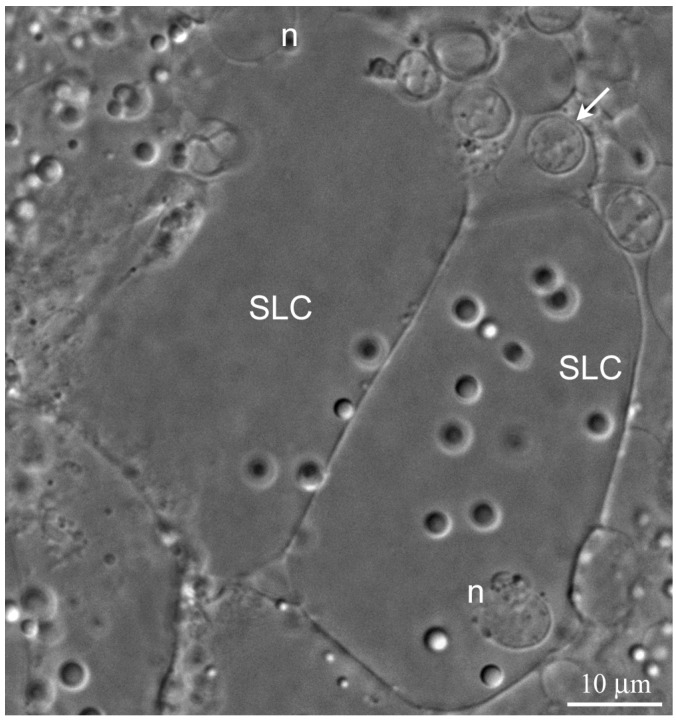
Young Sertoli-like cells (SLC) in the *L. saxatilis* testis wall. Leica DM 2500 microscope equipped with Nomarski optics (Differential Interference Contrast, DIC)-live imaging of a gonad surface. n, nucleus, arrow indicates a putative Sertoli-like cells (SLCs) progenitor.

**Figure 3 ijms-21-03792-f003:**
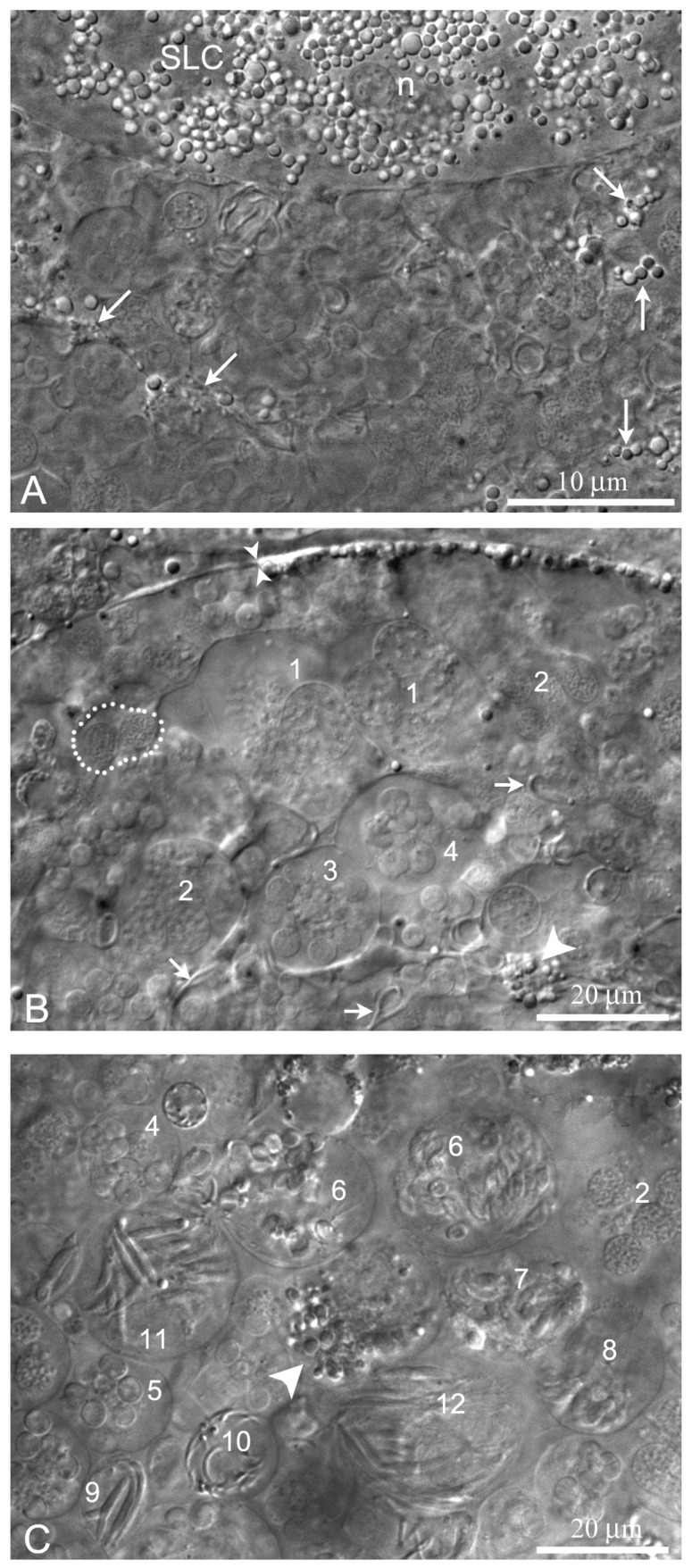
Spermatocysts at different stages of development in *L. saxatilis* testis lobules. High resolution DIC live imaging; (**A**) A fragment of mature Sertoli-like cell (SLC) in tangential optical section; n, SLC nucleus, arrows indicate basal cytoplasmic processes of the SLC that interleaf with developing spermatocysts in the lobule cavity; (**B**,**C**) cross-optical section through the apical (**B**) and inner part (**C**) of testicular lobule; small paired arrowheads marks the border between young and mature SLCs; big arrowheads point to basal cytoplasmic extrusions of SLCs; broken line encircles a goniablast cyst; short arrows indicate free spermatozoa; numbers mark multicellular cysts at the different stages of germ cell development, classified according [33]: 1, primary spermatocytes; 2, secondary spermatocytes; 3–12, spermatids during spermiogenesis: 3, stage 1 early spermatids; 4, stage 3 early spermatid; 5, stage 7 mid spermatids; 6, stage 8 mid spermatids; 7, stage 9 elongated mid spermatids; 8, stage 10 elongated mid spermatids; 9, stage 13 late spermatids; 10, stage 14 late spermatids; 11, stage 15 late spermatids; 12, stage 16 late spermatids.

**Figure 4 ijms-21-03792-f004:**
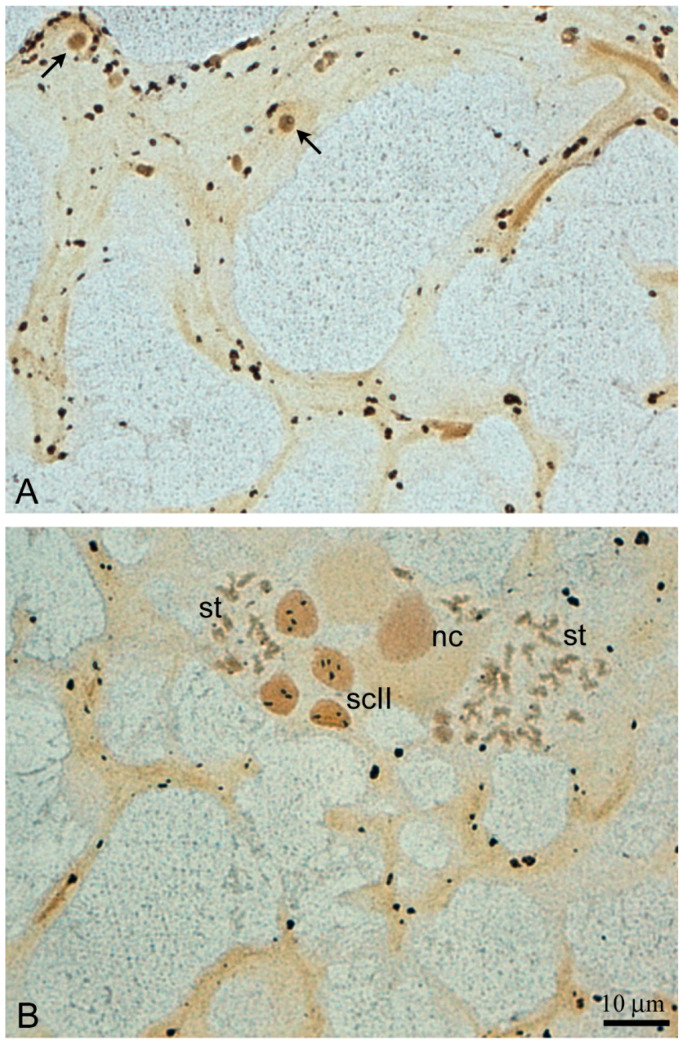
Silver-stained spreads of the lobule apex after spermatozoa migration from the lumen; (**A**) An anastomosed remnant of the meshwork formed by the cytoplasmic processes of Sertoli-like cells; arrows indicate the nucleus of the cells; (**B**) retarding late spermatid cysts (st) and secondary spermatocyte cyst (scII) with a polyploid nucleus of the cyst cell (nc).

**Figure 5 ijms-21-03792-f005:**
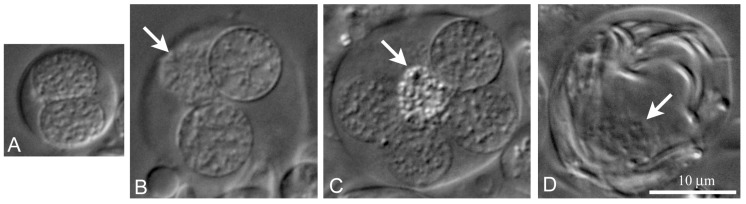
The first line of cyst differentiation in *L. saxatilis* testes. DIC live imaging. (**A**) a goniablast cyst; (**B**) cyst with spermatogonia; (**C**) cyst with primary spermatocytes; (**D**) cyst with late spermatids; arrows indicate the nucleus of somatic cyst cell.

**Figure 6 ijms-21-03792-f006:**
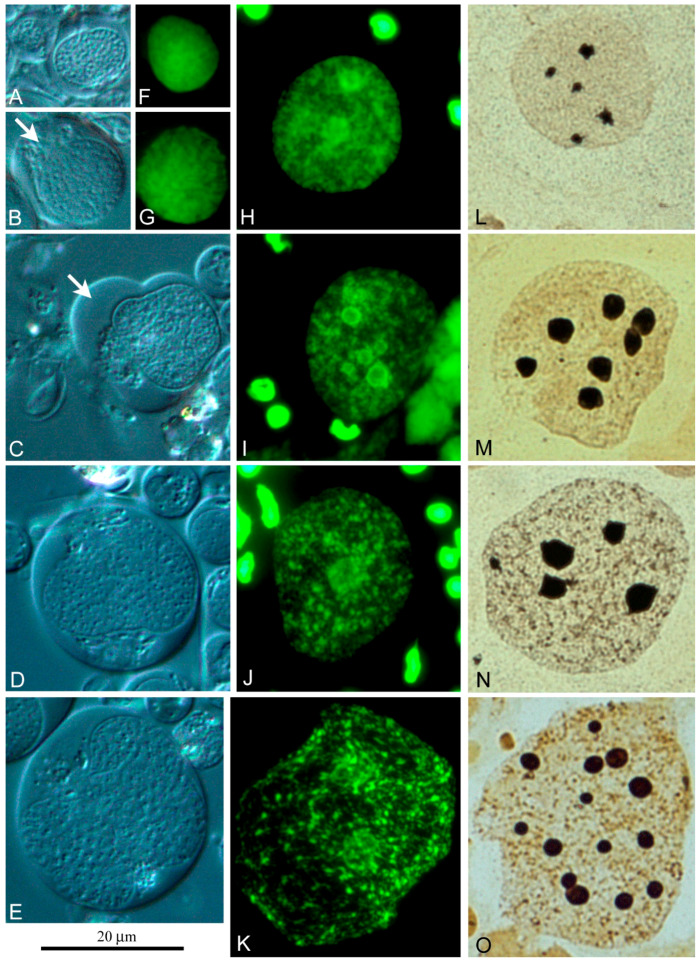
Free cyst cells with a different level of endoreduplication; (**A**–**E**) DIC live imaging; (**F**–**O**) Diploid and polyploid nuclei of free cyst cells after chromomycin A_3_ (**F**–**K**) and AgNOR staining (**L**–**O**); arrows in (**B**,**C**) indicate lobopodia of free cyst cells.

**Figure 7 ijms-21-03792-f007:**
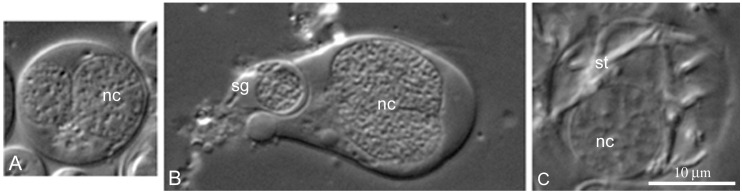
The second line of cyst differentiation in *L. saxatilis* testes. DIC live imaging: (**A**) spermatogonial cyst with polyploid somatic cyst cell; (**B**) polyploid free cyst cell capturing free spermatogonium (sg); (**C**) late spermatid cyst with polyploid cyst cell; nc, cyst cell nucleus; st, spermatids.

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
