# Peer review of "New Data on Spermatogenic Cyst Formation and Cellular Composition of the Testis in a Marine Gastropod, Littorina saxatilis"

_ijms, 2020, doi:10.3390/ijms21113792_

Round 1

Reviewer 1 Report

The authors Sergei Iu. Demin et al.,  has been submitted a manuscript in Int. J. Mol. Sci,  ID: 803849 with title “New data on spermatogenic cyst formation and cellular composition of the testis in a marine gastropod,  Littorina saxatilis”.

-General comments:

The authors if possible, should carry out the histological staining (for example hemalum-eosin), was to better highlight the edges of the cysts and to better highlight also the chromatin of the germ cells. Because, with the techniques used it is not possible to see the individual cysts and their distribution: Furthermore, it is difficult to understand the density of the chromatin.

In figure 1 The authors should also indicate the hepatopancreas with some symbols.

-Minor revision:

line 45 replace acystic with tubular organization. Do it for the whole text.

line 53 the authors should specify that the germ cells are distributed within the seminiferous tubules.

line 109 the authors should also describe another mollusk such as Mytilus galloprovincialis, in which spermatogenesis has been demonstrated take place within follicles that present an organization very similar to vertebrate seminiferous tubules. (Reference: Prisco, M.; Agnese, M.; De Marino A.; Andreuccetti, P.; Rosati L. Spermatogenic cycle and steroidogenic  control of spermatogenesis in Mytilus galloprovincialis collected in the bay of Naples. Anat. Rec. (Hoboken)  2017, 300, 1881–1894.

line 129 authors should correct the word hrpatopancreas with hepatopancreas.

I believe that the paper  would be suitable for pubblication in Int. J. Mol. Sci,  after the minor point suggested.

Minor point to be improved.

In the reference lack the n. 93 presente in paper in section (miRNA interaction prediction), please check the correspondence.

Author Response

We thank the referee for reviewing of our manuscript and his/her positive criticism, and valuable comments. Please see the attachment for answers.

Reviewer 2 Report

The English language is very poor has toe be edited by the native speaker 

However there is a problem with the description and possibly authors' understanding what is the cyst cell. The somatic cyst cells can not form the germ cells, these are two independent cell lines of different origin, authors make scientifically incorrect statements all over the text. The manuscript has to be completely rewritten and scientific errors corrected. I marked some  of the errors in yellow color.

In addition author should add to the introduction a paragraph why the snails are important from epidemiological point of view- they are the hosts for many zoonotic diseases

Author Response

(The authors gave the same response as above.)

Reviewer 3 Report

This paper describes two different testicular structures underlying the spermatogenetic process in the molluscs Littorina saxatilis.

This is an interesting study which adds a new contribution to the knowledge of spermatogenesis of invertebrates and beyond.

I have some comments:

Although this peculiar spermatogenic process is the aim of this work, authors should better introduce the general concepts of spermatogenesis. Authors describe in detail the cystic spermatogenesis but very shortly the sequential events of general spermatogenesis and especially the role of meiosis and the significance of haploid cells and so on. To expand this description (lines 34-39 and 74-75) may be beneficial for not expert readership.

Line 51 “clonally” do they mean by mitosis?

Some parts (as line 102-107) seem to be not appropriate for an introduction, better to move in the discussion.

The similarities with Drosophila are interesting but also questionable from an evolutionary point of view. Authors must better discuss the importance of their findings by comparing two species which are evolutionarily distant from each other. (lines 114-123). This is also in the case of Sertoli- like cells which are strictly related to mammalian reproduction rather than to marine invertebrates.

Line 133 explains how authors can be sure to have not induced damage.

Figure 3 is important but it is difficult to well visualize the different stages. If possible, put some inserts reporting higher magnifications.

Generally, this work is a morphological description of a double modality in the spermatogenic process. An effort should be made to interpret these data to provide a possible physiological rationale associated with each process.

Correct typo on line 129

The characterization of specific lobular-cystic type in this work is a novel finding of species-specific structures in the male reproductive system of gastropods. The work is well designed; the figures are convincing. Furthermore, the authors are expert in the field and this work is consistent with previous publications in the same field.

Author Response

(The authors gave the same response as above.)

Round 2

Reviewer 2 Report

I insist on including the importance of the snails for the human health, it will improve the value of the manuscript

Author Response

We thank the referee for his/her interesting suggestion.

Please see attachment for answer.
